# Decrease in RNase HII and Accumulation of lncRNAs/DNA Hybrids: A Causal Implication in Psoriasis?

**DOI:** 10.3390/biom12030368

**Published:** 2022-02-25

**Authors:** Ecmel Mehmetbeyoglu, Leila Kianmehr, Murat Borlu, Zeynep Yilmaz, Seyma Basar Kılıc, Hassan Rajabi-Maham, Serpil Taheri, Minoo Rassoulzadegan

**Affiliations:** 1Betul Ziya Eren Genome and Stem Cell Center, Erciyes University, 38280 Kayseri, Turkey; ecmel.mbeyoglu@gmail.com (E.M.); zeynepyilmaz_55@hotmail.com (Z.Y.); 2Department of Medical Biology, Medical Faculty, Erciyes University, 38280 Kayseri, Turkey; 3Animal Sciences and Marine Biology Department, Faculty of Life Sciences and Biotechnology, Shahid Beheshti University, Tehran 1983963411, Iran; h_rajabi@sbu.ac.ir (H.R.-M.); kianmehr96@gmail.com (L.K.); 4Dermatology and Venereology Department, Medical School, Erciyes University, 38280 Kayseri, Turkey; muratborlu@erciyes.edu.tr; 5Dermatology and Venereology Department, Training and Research Hospital, Aksaray University, 68000 Aksaray, Turkey; drseymabasar@gmail.com; 6INSERM-CNRS, Université de Nice, 06000 Nice, France

**Keywords:** psoriasis, non-coding RNAs, telomere length, RNA–DNA hybrid, TERRA, RNase HII, centromeres, epigenetics, cancer

## Abstract

Functional long non-coding RNAs (lncRNAs) have been in the limelight in aging research because short telomeres are associated with higher levels of TERRA (Telomeric Repeat containing RNA). The genomic instability, which leads to short telomeres, is a mechanism observed in cell aging and in a class of cancer cells. Psoriasis, a skin disease, is a disorder of epidermal keratinocytes, with altered telomerase activity. Research on the fraction of nascent RNAs in hybrid with DNA offers avenues for new strategies. Skin and blood samples from patients were fractionated to obtain the RNA associated with DNA as a R-loop structure. The higher amount of TERRA levels attached with each chromosome end was found with psoriasis patients in blood and skin. In addition to telomeric TERRA, we evidenced accumulation of others non-coding RNA, such as non-telomeric TERRA and centromeric transcripts. Increased levels of non-coding RNAs attached to DNA correlates with a decreased in Ribonuclease HII (*RNase-HII*) transcript which means that overall unresolved DNA–RNA hybrids can ultimately weaken DNA and cause skin lesions. Since the genome is actively transcribed, cellular *RNase-HII* is essential for removing RNA from the DNA–RNA hybrid in controls of genome stability and epigenome shaping and can be used as a causal prognostic marker in patients with psoriasis.

## 1. Introduction

Psoriasis is a chronic, recurrent inflammatory disease of the skin characterized by abnormal proliferation of keratinocytes, vascular hyperplasia, and infiltration of inflammatory cells into the dermis and epidermis [1,2]. It is generally reported that the central pathogenesis of psoriasis is the dysfunction of T lymphocytes affected by complex interactions between genetic and environmental factors, such as trauma, infections, stress, medications, smoking, and alcohol consumption. The disease affects both men and women [3]. The uncertain etiology of psoriasis characterized by scaly erythematous plaques that can cause significant physical and psychological distress and affect approximately 125 million people worldwide [4,5,6]. The localization of the skin lesions is determined according to the traumatic triggers, the most frequent local lesions are on the knee, the elbow, and the scalp [7]. The abundance of reported familial cases suggests genetics and epigenetics predisposition [8].

Keratinocytes normally proliferate over a period of 40 days, but in patients with psoriasis their proliferation accelerates to only 4–8 days [1]. For many years, it has been debated whether dermal inflammation or epidermal proliferation is the cause of abnormal keratinocyte proliferation in the pathogenesis [9,10]. Acceleration in cell division is also known to permit a higher error risk during replication. Immune-mediated inflammatory diseases (IMID) collectively describe a group of unrelated disorders that share common inflammatory pathways [11], such as inflammatory bowel disease (IBD), psoriasis (PS), rheumatoid arthritis (RA), spondylarthritis, and uveitis, which belong to the group of age-related non-communicable diseases (NCDs) [11,12,13,14]. Chronic inflammation, oxidative stress, and alterations in telomere length (TL) have been shown to be involved in various age-related NCDs [15,16,17,18].

Telomeres consist of the double strand TTAGGG repeating units of DNA sequences associated with six telomere-specific proteins called Shelterin complex, three of them—Telomere Repeat binding Factor 1 (TRF1), Telomere Repeat binding Factor 2 (TRF2), and Protection of telomere 1 (POT1)—recognize repeated motifs to protect each chromosome ends [16,17,19]. Additionally, TRF1- and TRF2-Telomere Interacting Nuclear Protein 2 (TIN2), Adrenocortical Dysplasia Protein Homolog (TPP1, also known as ACD), and Repressor/Activator Protein 1 (Rap1) proteins allow telomeres distinction from any DNA damage sites. Recently, functional long non coding RNAs (lncRNA), Telomeric Repeat- containing RNAs (TERRA) has been described with UUAGGG repeats which are transcribed directly from the end of each chromosome [20] and hybridize to the leading strand and form a DNA–RNA R-loop [21,22]. In addition, the remaining DNA single-stranded lagging 3′ end invades the double-stranded telomeric helix, forming a T-loop structure [22,23,24]. A specific reverse transcriptase complex (TERT) carrying its integral RNA template (TERC) adds telomeric repeats to the 3′ ends of chromosomes, to maintain telomeric sequences [25,26]. Telomerase, is active at the early stage of development and in adult stem cells but mainly inactive in human somatic cells (except in lymphocytes) [27]. Its activation is a marker of rapid cell division in many cancer cells (85–90% of all human cancers) [28,29]. Altered TL has been associated with aging and aging-related diseases [30]. Telomere dysfunctions due to breaking or shortening during lifespan ultimately leads to cell senescence with chromosomal instability [22,31,32].

The rate of telomere shortening may be accelerated by inflammation and oxidative stress in vitro and participate in the aging process [33]. Wu et al. [34,35] demonstrated an increase in telomerase activity associated with a reduction in TL in T lymphocytes compared to healthy controls [26] and it was also confirmed in Peripheral Blood Mononuclear Cells (PBMCs) [36] in patients with psoriasis.

It is proposed that TERRA regulates the structure of telomeric chromatin [21,32], but the mechanisms by which TERRA functions in disease development are poorly defined. It is reported that TERRA is preferentially recruited to short telomeres [15,18,24,32] increasing the formation of hybrid DNA–RNA telomeric structures (R-loop) with the higher levels of TERRA triggering telomere fragility. Since genome is actively transcribed in this study, we focused on the nascent transcripts of lncRNAs in association with genome as an epigenome labeling.

We propose a model where higher levels of lncRNAs held in hybrid with DNA, including the ends of chromosomes, indicate epigenetic changes in the genome and could link to predisposition to the diseases. We recently reported [37] in normal cells that a fraction of nascent transcribed TERRA is strongly maintained as a DNA–RNA hybrid in the telomeres from which they are derived. Here, we applied the same protocol to dissect and assess the levels of lncRNA molecule fractions, which are in hybrid with DNA in lesional (LA) and non-lesional (NL) skin and blood samples of psoriasis patients. Our results here reveal the detection of a general increased level of lncRNAs in R-loop with genome which correlates with decreased levels of Ribonuclease HII (*RNase HII*) transcripts in the skin of the psoriasis patients. *RNase HII* present in all organisms is required to cleave RNA primers of RNA–DNA hybrid during DNA replication [38,39]. *RNase HII* knockout mice are not viable, and mutations in the human genes cause uncleaved RNA–DNA hybrids with Aicardi–Goutières Syndrome, a severe inheritable neurodevelopmental disorder [40]. Overall, detection of an increased level of DNA–RNA hybrid R-loop association in psoriasis due to decreased RNase HII levels, providing a definite causal effect of higher TERRA to shorten telomeres length during the progression of psoriasis.

Moreover, higher levels of DNA interactions with RNAs (strengthens) reveal epigenetic marks that distinguish the epigenome of patients with psoriasis from the epigenome of unaffected individuals.

## 2. Materials and Methods

Patients diagnosed with psoriasis and healthy volunteers admitted to the Skin and Venereal Dermatology unit of the Faculty of Medicine at Erciyes University were included in this study after approval by the university’s Human Ethics Committee: Erciyes University with decision no 17/002.

### 2.1. Patient Selection

Skin biopsy samples were taken from the area with (LA) and without lesions (NL) from patients with psoriasis, who agreed to participate in the study admitted to the Department of Dermatology of the Faculty of Medicine at Erciyes University. The research method of the patients included in the study was prospective, and they had a clinical diagnosis of psoriasis. After calculating the psoriasis area severity index (PASI) [41] value of patients who did not receive systemic therapy in the past four weeks, patients with a PASI value greater than 10 were included in the study.

A total of 35 people over the age of 18 years old were included in the study, including 20 patients diagnosed with chronic plaque-type psoriasis and 15 healthy volunteers who were consistent with age and gender as a control group. The psoriasis patient group consisted of 10 women and 10 men patients and the healthy group consisted of 8 women and 7 men. The mean age of the control groups was 42.6 and 39.5 years, respectively. There is no statistically significant difference between the groups in terms of age and sex (*p* > 0.05).

Patients younger than 18 years old, patients with an inflammatory/autoimmune or chronic disease other than psoriasis, and patients who received topical therapy for psoriasis in the last two weeks or systemic therapy in the last four weeks were not included in the study.

### 2.2. Taking Skin Biopsy Samples

Skin biopsy samples of LA and NL areas of 20 psoriasis patients included in the study. Tissue samples were taken from the NL skin adjacent to this lesion and from the normal skin area behind the knee (popliteal fossa) of the volunteers in the control group, with a 2 mm punch tool after local anesthesia was applied under the skin, and completely sterile conditions were provided. Tissue samples from patients and controls were immediately delivered to the Genome and Stem Cell Center (GenKok) Genome Unit. Half of the samples were used for DNA isolation and DNA–RNA hybrid determination to assess TL and hybrid TERRA amount, and the other half for RNA isolation to determine the total expression of TERRA. The isolated samples were stored at −80 °C until used.

### 2.3. DNA and DNA–RNA Hybrid Isolation

DNA, RNA, and DNA–RNA hybrid isolation was performed in accordance with the manufacturer’s protocol, using the ZR-Duet DNA–RNA MiniPrep Plus Kit (ZYMO Research (www.zymoresearch.com (accessed on 24 January 2022) ZR-Duet ^TM^ DNA–RNA MiniPrep Plus Cat No: D7003, Irvine, CA, USA)), from tissue samples. The concentrations and purity of DNA, RNA, and DNA–RNA hybrid samples were measured in the Biotech Biospec-Nanodevice.

### 2.4. Determination of Telomere Length

Telomere lengths from obtained DNA samples were determined in a quantitative real-time Light Cycler 480 PCR (Roche, Mannheim, Germany) device using appropriate primers (Tel F:CGGTTTGTTTGGGTTTGGGTTTGGGTTTGGGTTTGGGT and Tel R:GGCTTGCCTTACCCTTACCCTTACCCTTACCCTTACCCT) and standards [42]. Telomere lengths were determined using the house-keeping gene (36B4F: CAGCAAGTGGGAAGGTGTAATCC and 36B4R: CCCATTCTATCATCAACGGGTACA) and then proportioning the obtained Ct values to the 36B4 reference gene.

### 2.5. Quantitative Real-Time PCR (RT-qPCR)

In order to determine the TERRA expression level from total RNA and DNA–RNA hybrid samples obtained from the patient and control groups; cDNA synthesis was performed using the EvoScript Reverse Transcriptase cDNA synthesis (Cat No: 07912374001, Roche, Mannheim, Germany) kit. The cDNA samples of the total RNA and DNA–RNA hybrids of the patient and control groups obtained were determined in the Roche Light Cycler^®^ 480 Real Time PCR (Mannheim, Germany) device using previously reported [43] primers RNase HII (F: GACCCTATTGGAGAGCGAGC and R: TATTTGACCCGCCCAAGCAT), Rad51 (F: GGCCATTAGCCCTTCACCAT and R: TCTGCAAGTGGGACTTTCCT) and GAPDH as the house-keeping gene. All samples were normalized using the data $2^{-\Delta Ct}$ method after a double study [44].

### 2.6. Determination of Genes Expression Levels

RT-qPCR experiments were performed using the high-throughput Light Cycler 480 II Real-Time PCR system (Roche, Germany, Mannheim). In order to determine the genes expression levels of total RNA samples obtained from patient and control groups, cDNA synthesis was performed using the EvoScript Reverse Transcriptase cDNA synthesis (Cat No: 07912374001, Roche, Mannheim, Germany). cDNAs were diluted with nuclease free water in 1:5 ratio. SYBR Green Master Mix (Cat No: 04707516001, Roche, Mannheim, Germany,) was used to determine mRNA expression levels of *Rad51* and *RNAse HII* genes in total RNA of tissues and blood samples.

The reaction mix was prepared according to the manufacturer’s instructions: 10 µL of 2× SYBR Green mix, 5 µL of nuclease free water and 0.5 µL of 10 pmol primer assays (*Rad51* F: GGCCATTAGCCCTTCACCAT and Rad 51 R: TCTGCAAGTGGGACTTTCCT; *RNase HII* F: GACCCTATTGGAGAGCGAGC and RNase HII R: TATTTGACCCGCCCAAGCAT) were mixed and dispensed into 96 multi-well plates. The total volume was completed to 20 µL by adding 4 µL of cDNA on it. Thermal cycling conditions consisted of an initial denaturation step at 95 °C for 5 min, followed by 40 cycles of 94 °C for 20 s, 60 °C for 20 s, 72 °C for 45 sn and 95 °C for 15 s, and, finally, melting curve was performed at 67 °C for 01 sn (melting curve) and cooled at 40 °C for 30 s. Human beta-actin (*ACTB* F: CTCGCCTTTGCCGATCC and ACTB R: TCTCCATGTCGTCCCAGTTG) was used as the reference gene. Changes in gene expression were determined using the 2^−ΔΔCt^ method of relative quantification in all groups.

### 2.7. Statistical Analysis

The suitability of the data for normal distribution was evaluated by histogram, q-q graphs, and Shapiro–Wilk test, and Variance homogeneity with Levene’s test. A two-sample *t*-test was applied to compare the differences in telomere lengths and TERRA expression of lesional and non-lesional tissues among patients. A One-Way ANOVA test was used to multiply compare Telomere lengths and TERRA expressions between patients and the control group. The relationships between quantitative data were evaluated by Spearman correlation analysis. Data were treated using Graphpad Prism (version 8.0.1, San Diego, CA, USA) software. P values were considered statistically significant as <0.05. One-way ANOVA analysis of peak annotation data were carried out using the SPSS Statistics for Windows version 19.0 (IBM Corp. Released 2010. IBM SPSS Statistics for Windows, Version 19.0.Armonk, NY: IBM Corp, Armonk, NY, USA). The mean of all the sample groups psoriasis patients and healthy controls data were compared using ANOVA followed by Duncan’s multiple comparison tests. The data were presented as means ± SD.

### 2.8. RNA Library and Sequencing

DNA-associated RNA molecules were purified from nine skin sample biopsies (three Control (C); three lesional (LA); three non-lesional (NL). After digestion of DNase, 10–100 ng of RNA were obtained.

Plateform Génomique Institut de Biologie – IBENS (ENS) Paris (Paris, France) performed libraries of all samples corresponding to DNA-associated RNA molecules and small RNA high-throughput sequencing (high-throughput sequencing on Illumina Hiseq 2500 or Illumina MiSeq (Paris, France)). Totally, 14 million (m) paired-end reads were mapped to unique sites in the human genome (1L), 20m (2L), 8m (1N), 17m (2N), 7m (2C), 8m (3C), and 10m (5C) per sample. Average length of the reads was about 75 nt. All the primary sequence characteristics, for sample libraries of the DNA-bound RNA sequences of psoriasis skin samples are summarized in Appendix A.

### 2.9. Sequence Analysis

FastQC version 0.11.7 was used to do some quality control on the RAW sequencing data. Illumina adapters sequences removed by cutadapt v1.16. Based on the FastQC results, trimming of bad-quality reads was performed using Trimmomatic version 0.36 (10 nucleotides were cropped from the 5′ end of each read, and trimmed bases with Phred score lower than 20 from heading and trailing of each read and the trimmed reads with a size less than 30 bp were removed) [45]. Then aligned to the reference assembly (GRCh38) using Hisat2 [46]. Quality of alignment assessed using plot-bamstats utilities of samtools (Li, 2011 #10323). Bam files converted to BigWig format using deep tools. Coverage with the following parameters: −of = bigwig − binSize 50 – normalize Using RPGC – effective Genome Size 2913022398. Visualization of normalized BigWig files was performed using web-based UCSC genome browser (http://genome.ucsc.edu, accessed date: 24 January 2022) [47,48].

### 2.10. HOMER Version 4.9 Was Used for Peak Finding

Genome-wide locations of TERRA repeats were found by aligning sequences with lengths of 24 and 48 nucleotides composed of four and copies copies of TERRA to the reference genome using bowtie2 using “−a” argument to report all alignment [33,37].

Statistically significant peaks of expression were identified using HOMER (10,000 size of region used for local filtering, four-fold over local region, Poisson *p*-value over local region <0.0001, false discovery rate (FDR) threshold <0.001).

### 2.11. Peak Finding

To identify significant peaks, we looked for genome-wide enriched peaks. This analysis aimed to efficiently visualize the peaks in terms of genomic region characteristics. The results are consistent between replicates, confirming that the technique is generally very reliable (see Figures 5–8). To find enriched peaks and regions associated with nearby genes and genomic features in the genome using Homer, “findPeaks –o auto” called out was used to perform peak calling and transcript identification analysis. Then to annotating regions in the genome “annotatePeaks.pl” called for performing peak annotation. FDR cutoff of 0.001 used for significant peaks identification. Annotated positions for different regions plotted. The percentage of significant peaks located in each genomic region is visualized. Annotated positions for exons, intergenic, intron, promoter-TSS, and TTS are based on the hg38 genomic features Peak annotation.

### 2.12. Statistical Analysis for Genomic Data

Because of the rheostat-type (non-normal) distributions and the limited number of patients, non-parametric methods were used, and a median value comparison was performed by Wilcoxon rank analysis. Statistical analyses were performed with an unpaired *t*-test with Welch’s corrections. Data are expressed as the median with *p*-values < 0.05 considered to be statistically significant. The standard error of the mean (SEM) represented as mean ± SD.

### 2.13. Data Access

To review GEO accession GSE188763: Go to https://www.ncbi.nlm.nih.gov/geo/query/acc.cgi?acc=GSE188763 (accessed on 24 January 2022). Enter token axyfuoyyjbyjlkt into the box.

## 3. Results

Twenty patients with chronic plaque psoriasis were included in the study. Ten of these patients were women and 10 were men. The healthy group consisted of 15 people, 8 women (53.33%) and 7 men (46.67%). Three sample groups of lesional (LA) and non-lesional (NL) psoriasis and healthy control were analyzed. The mean age of the control groups was 42.6 and 39.5 years, respectively. There is no statistically significant difference between the groups in terms of age and sex (*p* > 0.05). The mean of Psoriasis Area and Severity Index (PASI) of the patient group was 15.6.

### 3.1. Telomere Length Is Significantly Reduced in Lesional Tissue in Patients with Psoriasis

Clear differences were observed in Figure 1 between psoriasis lesional (PSO LA) and non-lesional (PSO NL) skin samples in terms of TL. TL was found to be significantly shorter in PSO LA compared to PSO NL tissue and the control group (*p* < 0.05). There was no significant difference in TL between PSO NL skin samples from patients with psoriasis and healthy controls (Figure 1).

Persistent DNA damage with disease progression can lead to gradual changes in chromatin structure and erosion of the regular epigenetic landscape, which can be particularly harmful in regions of constitutive heterochromatin. According to several reports [18,25,30,31,33,36] this potentially involves epigenetic alterations associated with aging in particular in the repetitive DNA sequences, including telomeric regions. We then tested the expression levels of lncRNA, such as TERRA, in the possible epigenetic alteration of TL.

### 3.2. A higher Level of TERRA Is Associated with Telomeres in Psoriasis Patients

We designed experimental strategies to determine whether short telomere in LA tissue in psoriasis is associated with increased TERRA levels.

Recently, we described a simplified method of detecting DNA–RNA hybrid regions in a complex genome. Unlike the often used DRIP-seq techniques (DNA–RNA immunoprecipitation and DNA strand sequencing) [49], it is not based on the recognition of the antigen or RNase H of the hybrid, but on a highly reproducible procedure to detect DNA–RNA hybrids with stable liaison that are maintained during the extraction protocol with TRIzol-chloroform [37,50]. DRIP assays carry more risk of artifacts due to immunoprecipitation steps and/or bias in antigen recognition. After extraction with TRIzol-chloroform the nucleic acids, are purified into two RNAs fractions, a small but reproducible amount of RNA is retained, along with high molecular weight genomic DNA, at the chloroform–water interface, while -free RNA molecules are found in the upper aqueous phase. The RNAs retained in DNA–RNA complexes were then recovered by chromatography on a “Zymo-SpinTM” column binding the DNA (Zymo-Research Corp Irvine CA, USA) followed by treatment with DNase and further purification with RNA strand analysis see Methods in contrast to DRIP with often DNA strand analysis. The routine free RNAs fraction was also purified from the aqueous phase after precipitation with isopropanol. Along with RT-qPCR, we examined the level of TERRA expression specifically transcribed from each chromosomal end on both fractions.

The aqueous phase extracted total RNAs (free fraction) from, both LA (lesional) and NL (non-lesional) skin samples from patients with psoriasis expressed TERRAs (chromosomes 1q, 2q, 7p, 9p, 10q, 13q, 15q, 17p, 18p, X_q_Y_q_, and X_p_Y_p_ see Appendix A and total TERRA Figure 2).

The relative levels of TERRA expression showed low or no significant difference between the three groups of total free RNAs fractions, but, in contrast, the DNA-bound fraction of TERRA profiles was significantly altered in the PSO LA and PSO NL skin tissue of affected patients with psoriasis. The difference is more significant in PSO LA (Figure 3).

The DNA-bound RNA hybrid (DRNA) obtained from PSO LA and PSO NL skin biopsy samples, compared to the control group shows a significant difference (*p* < 0.05) in the relative levels expression of TERRA maintained to the chromosomes 1q, 2q, 7p, 9p, 10q, 13q, 15q, 17p, 18p, XqYq, XpYp, and sub-telomeric regions (Appendix A).

Increases and decreases levels of TERRAs expression are associated with variable telomere length and can trigger uncontrolled cell division, genome instability, cell senescence and disease [15,43].

Taken together, a higher level of TERRA was found in LA and NL DRNAs in psoriasis skin samples. This higher level is already detected in the non-lesional samples of patients with psoriasis, which means that the increase of TERRA in the fraction of RNA bound to DNA precedes the shortening of chromosomes length in the skin. Thus, higher levels of TERRAs associated with the genome of psoriasis patients could indicate a source in lesion development.

In addition, DNA and DRNA were isolated from both the patient’s total blood cells and the healthy controls. Then, the length of the telomeres and the level of expression of TERRA were determined by RT-qPCR. According to our results, in blood cells from patients with PSO have a significantly shorter telomere length than the healthy controls. Expectedly, PSO patients with shorter telomere length have a significantly higher TERRA level attached to DNA (Figure 4). These results are consistent with the TL of the lesion tissue and the higher TERRA level results as DRNA.

### 3.3. Transcription Profiles of Non-Coding RNAs in Patients with Psoriasis: DNA-Associated RNA Fraction Analysis Reveals Higher Levels of R-Loop in the Genome

To get an overall view of LA and NL tissue RNA profiling, RNA-seq analysis of DNA associated RNA fraction (DRNA) was performed. Recently, we reported that with a conventional sperm RNA extraction protocol, unlike the free RNA fraction, a fraction made up of DNA–RNA hybrid molecules is tightly associated with the genome [37,50]. The DRNAs were recovered with the same way from skin samples of six healthy controls and six LA and NL samples of psoriasis patients.

To prepare the RNAs for the RNA-seq, the cutaneous cells were treated with proteinase K and then the DNA and RNAs fractions were recovered by column separation. The DNA retained on the columns was treated with DNase, and the RNA was released and further purified. The RNA fraction recovered from DNA is referred to as fraction D or DRNA for details see reference herein [37,50]. The DRNAs sequencing was performed by an external service (see Materials and Methods). To identify significant peaks, we looked for genome-wide enriched peaks. The results are consistent between replicates, confirming that the technique is generally very reliable.

We assessed RNA sequences from human psoriasis skin cells of Lesions (LA), Normal (NL) and Control (C) individuals. LA and NL biopsies were taken from the same individuals. We first focused on abundantly expressed TERRA transcripts. Interestingly, DRNA fraction from human somatic cells shows telomeric TERRA hybrids. Figure 5 shows RNA-seq signals over TERRA sequences at p- and q-arm telomeric regions of several human chromosomes. Each track shows a different sample (red colors are DNA–bound from lesions, blue colors from normal, and green from control samples). LA is DNA-bound RNA samples from lesions, NL is from normal, and C are controls). The heights of the signals show group auto-scaled expression level of normalized BigWig files over each genomic region of different chromosomes using UCSC genome browser.

#### 3.3.1. Telomeric Regions Retain More TERRA in Patients with Psoriasis

To view genome-wide TERRA loci throughout the different chromosomes, DNA–RNA hybrid density tracks along with TERRA loci (shown as a green bar) have been visualized on the ends of two arms of most of human chromosomes (Figure 5). TERRA loci are minimum four consecutive sequences TTACCC. These tracks display read density over telomeric TERRA regions based on hg38 genome assembly that reveal TERRA expression signal in normalized bigwig files. Each graph-based track is configured to highlight different samples (LA in red, NL in blue, and controls are in green colors) value (scales shown at the left of each browser). Signal tracks of LA, NL and controls revealed expression signal at TERRA regions at p-arm or q-arm or both end of different human chromosomes (1p, 3p, 4p, 5p, 11p, 12p, 17p, and 18p), and (1q, 2q, 3q, 4q,10q, 12q, 15q, 17q, 20q, 22q, and Xq). TERRA signals are present at both ends of chromosomes 1, 3, 4, 12, and 17. The comparative analysis of fully sequenced sub-telomeres of human chromosomes; 1p, 3p, 4p, 5p, 11p, 12p, 17p, 18p, 1q, 2q, 3q, 4q, 10q, 12q, 15q, 17q, 20q, 22q, and Xq has revealed a common structure, in which the proximal and distal sub-telomeric domains are separated by TTAGGG repeats. TERRA repeats are indicated within upstream and downstream regions in average 1–50 kb on every chromosome ends to show a range of transcripts proximity.

RNA-seq expression profiling of LA, NL, and control samples shows that TERRA levels signals are relatively higher in LA and NL than from controls (Figure 5).

This observation also confirms the RT-qPCR results according to which the DRNA level of TERRA in the LA and NL samples groups is much higher than the control samples groups.

Finally, Costa et al., also reported a novel gene family, indicating that an ancestral gene, originated as a rearranged portion of the primate DEAD/H-Box Helicase 11 (DDX11) gene, and propagated along with many sub-telomeric locations of human chromosomes [51]. Sub-telomeric transcripts of the DNA-bound RNA molecules may also occurs from every chromosome in human somatic cells as well as germ cells [37] data not shown.

#### 3.3.2. Non-Telomeric TERRA Regions Retain More RNAs

Moreover, we also reported in our previous study that in addition to telomeric TERRA regions there are some non-telomeric regions in sperm DRNA fractions [37]. Non-telomeric region of TERRA repeats are shown on chromosomes 2, 10, 18, and 22 (Figure 6).

#### 3.3.3. More lncRNA Expression Signals Are Observed in Non-Telomeric Regions of the Lesional and Non-Lesional Skin of Patients with Psoriasis

Annotated positions of exons, intergenic, intron, promoter-TSS (Transcription Start Site), and TTS (Transcription Termination Site) were based on the hg38 assembly genomic features which shows that most of the peaks in skin DRNA samples fall into enhancer regions (intergenic and intronic regions) compared to others. The general transcripts of the regions: exon, promoter-TSS, and TTS are perfectly comparable levels in all samples (Figure 7). Thus, more variations are observed in intergenic and intronic regions.

On the other hand, in accordance with the RT-qPCR technique results, higher expression values are also observed in non-lesional samples from psoriasis patients by the report to the control. But, the lesional samples profiling is more complex. More analysis is required to compare other genomic regions, such as ribosomal, etc., that can be considered for further studies.

#### 3.3.4. Higher Levels of Transcripts from Centromeric Regions Are Retained on the DNA

Centromeric regions (see below) also accumulate more DNA-bound RNA compared to controls not only in lesion samples but also in the healthy samples from patients with psoriasis (Figure 8).

It is accepted that RNAs are associated with centromeric chromatin. In humans, R-loops are detected at centromeres in mitosis; and an R-loop-driven signaling pathway promotes faithful chromosome segregation and genome stability [52]. It is a common feature that the centromeric regions are desertic in the gene [53,54]. In our data, centromeric RNAs fall in gene-free regions. As shown in Figure 8, RNAs signals are at flanking pericentric heterochromatic of centromeres.

With deep-sequencing analysis, we detect signals at the centromeric regions of chromosomes 4, 5, 12, 13, 20, 21, and Y, of all DNA-bound RNA samples with highly significant enriched repeat sequences of CCATT/GGTAA (Figure 8). Most of them are located at up-stream regions or promoters so they may have a role in regulating gene expression. Interestingly, centromere read density signals of chromosome 13 are overlapped with the Family with Sequence Similarity 230 Member C (FAM230C) lncRNA. On the centromeric region of chromosome 20, two enriched signals were found: one is 79 bp nearby the FSHD Region Gene 1 Family Member C (FRG1Cp) pseudogene, and, in another one, the repetitive CCATT/GGTAA sequences are FSHD Region Gene 1 Family Member D (FRG1DP, another pseudogene). Centromeric regions sequences of DRNAs are overlapped to simple repeats sequences as shown using the UCSC genome browser.

These observations are in accordance with previous reports that the centromeric sequences have a repetitive nature (alpha-satellite tandem repeats) [55]. The centromeric expression signals of RNA-seq data overlap with repeat markers including simple repeats (micro-satellites, Figure 8). It has been determined that despite the repetitive sequences, various types of centromeric RNAs play an important role in centromeres as centromeric regions are transcriptionally active, and transcripts are processed, including small RNAs, lncRNAs, circRNAs, and hybrids DNA–RNA, which are associated with centromeric proteins and pericentric heterochromatin [55]. It has been shown that the accumulation of R-loops at the centromeric chromatin can alter the integrity of the genome in yeast and mammals [55]. We have also noticed that the DRNA fractions are visualized in the centromere regions of many chromosomes with higher levels in LA and NL than healthy controls. When centromeric regions are compared in Figure 8, centromeric transcripts are more abundant in LA tissue relative to NL and control tissue. In addition, centromeric RNA is relatively high in NL tissue compared to controls.

Our hypothesis was that the accumulation of DNA–RNA hybrids more in patients with psoriasis may lead to chromosomal instability, which can result in the development of lesions in patients with psoriasis. Evaluation of the skin genome associated transcripts here by deep sequencing revealed that the amount of non-coding RNAs such as TERRA, non-telomeric TERRA, from intron and transcripts from centromeres are retained with DNA and is generally increased in psoriasis samples compared to controls (Figure 6, Figure 7 and Figure 8).

### 3.4. Decreased Transcription Levels of Ribonuclease HII in Patients with Psoriasis

Here we report an increase in the level of several lncRNAs associated with DNA, which means an increase in the non-physiological formation of the R-loop that could trigger genome fragility. DNA repair protein RAD51 and its partner Breast Cancer 2 (BRCA2) interact with TERRA and promote the association of TERRA with short telomeres [25]. On the other hand, factors in particular, Ribonuclease H and TRF1 oppose the formation of the R-loop [22,39,56,57,58,59]. These proteins play an important role in the physiology of telomeres. We then used the RT-qPCR technique to assess the transcript levels of *Rad51* and of Ribonuclease HII (*RNase HII*) in patients with psoriasis. The results are shown in Figure 9. While the decrease in the level of transcripts in *RNase HII* is highly significant especially in the LA tissues of psoriasis patients, in contrast the levels of Rad51 transcripts do not seem to increase. According to these results, correlation analysis was done and the positive and moderate relation was detected between telomere length and *RNase HII* expression in PSO lesional tissue (*p* = 0.001 rho = 0.688). Taken together, these results indicate that an increase in DNA-associated lncRNAs levels could be due to the decreased levels of *RNase HII* transcription in skin tissues of patients with psoriasis. In addition, we determined the level of expression of the *RAD51 and RNase HII* transcripts in blood samples from PSO patients and controls. Patients with PSO have a lower expression level of *RAD51* and *RNase HII* transcripts compared to healthy control in blood; but these results are not significant.

## 4. Discussion

Here, in clinically defined psoriasis, we have revealed, with molecular tools and genomic knowledge, modification of the retention levels of lncRNAs on DNA. We propose causal epigenomic mechanisms that add to the already described clinical pathways of phenotype and cytokines. The aim of this study was to search epigenetic signals in patients with psoriasis. Our hypothesis is that altered epigenetic signals are transcripts such as TERRA due to shorter telomere observed in psoriasis lesion. In fact, our results show a higher amount of TERRA molecules attached to the telomeres (psoriasis genome). Thus, similar to TERRA, the RNAs mediated epigenetics signals must be in this disease strongly attached to the DNA (genome). To achieve our goals, we are measuring the level of lncRNAs such as TERRA associated with telomeres and others long non-coding RNAs (non-telomeric TERRA and centromeric) in the genome of patients with psoriasis. We also know that psoriasis cells develop short telomeres after lesions, and must, therefore, contain one or more epigenetic changes that make them susceptible to lesions under certain conditions. Our results indicate that, indeed, lncRNAs, such as TERRA, centromeric, and others, RNAs are retained in R-loop at higher levels on the genome of patients with psoriasis than controls. The results demonstrated more than expected, an increase in overall R-loops in the genome indicates changes in the activity for removal of the RNA fraction of DNA–RNA hybrids. Indeed, the cellular transcripts levels of RNase HII essential for the degradation of RNA templets are reduced in patients with psoriasis. Here, we propose a way to predict the severity and occurrence of the lesion, or possibly to be used as a source of causation and development of inflammation due to the endogenous epigenetic changes.

### 4.1. Telomere Length Is Shorter in Patients with Psoriais

As it is known, telomeres, which are ribonucleoprotein structures at the ends of linear chromosomes, prevent the natural ends of chromosomes from being recognized as DNA breaks and can lead to inappropriate repair activities, such as telomere dysfunction, homologous recombination (HR), and non-homologous junction. Due to incomplete DNA replication and nucleolytic degradation, telomeres shorten with each replication cycle, ultimately leading to cell death, known as replicative senescence or apoptosis [24].

In psoriasis, keratinocytes proliferate faster than in normal people. This situation is caused by the formation of keratinized tissue in patients with psoriasis. In fact, it is due to the increased rate of proliferation and accelerated death of keratinocytes resulting from this rapid cell proliferation, the barrier skin deteriorates, and inflammation occurs in patients. This indicates that either certain localized factors cause a further shortening of the telomere length in the lesion tissue, or because the keratinocytes in the lesioned area continue to divide faster than in normal skin.

### 4.2. TERRA Profiles Vary in Patients with Psoriasis

In this study, we used analysis of TERRA retention profiles on DNA of the skin as marker. Our studies revealed that the increased level of RNA attachment to telomeres correlated with the state of patients with psoriasis. We evidenced that the amount of TERRA associated with DNA was higher in patients with psoriasis even in non-lesional tissues (skin and blood) than in healthy controls. Interestingly, the overall levels of TERRA are not much changed but regulation of TERRA levels depends on the location of TERRA in loci associated with chromosomes. This highlights alterations in pathways that protect actively transcribed regions from genotoxic stresses. Here, it is observed localized epigenetic regulatory changes with development of TL abnormalities implicated in disease progression. In particular, damaged skins progress to telomeric shortening. These results supported the hypothesis that increasing the level of TERRA in the R-loop structure compromises the long-term maintenance of TL and cell survival. These keratinocytes cells from skins of psoriasis patients do not even over-express TERRA overall compared to the control, but are more retained in the genome at the telomeres. The amount of TERRA retained in the chromosome is already high in patients with psoriasis, and how TL is controlled in normal tissues compared to scarred spots requires further study.

It has been reported that the lesions promote and accelerate cell proliferation in culture experiments. The downstream signaling pathway of cell growth acceleration has not yet been clarified, but several reports have shown that short telomeres play a role in triggering cell replication and cancer [15,16,24]. Here we report that higher TERRA retention was associated with psoriasis patients suggesting that this could be a potential marker of determinant effects. Interestingly, TERRA telomere retention is revealed in the same patient skin with or without lesions. These results are consistent with previous reports by Wu et al. [35], on higher telomerase activity in patient blood cells, and now with increasing TERRA levels in blood and skin could be a coherent epigenetic event for early detection of psoriasis. Thus, studies on TERRA retention on telomeres and others lncRNA on the genome might help to follow-up in determination of causality, mechanisms, and trans-generational studies.

We observed a relationship between TERRA retention and disease progression, with decreased levels of *RNase HII* transcripts. The detailed mechanisms by which skin cells negatively regulate *RNase HII* transcription to let more TERRA in telomeres remain unknown. Further studies are needed to clarify the relationship between TERRA and intercellular signaling. In addition, TERRAs are not the only lncRNA retained with DNA in patients with psoriasis, in fact, centromeric regions and other regions also exhibit higher levels of RNA retention on DNA. This suggests an alteration of the general pathways of resolution of hybrid DNA–RNA structures due to robust down-regulation of *RNase HII*.

### 4.3. Dysregulation of R-Loop Structures Reveals Genome Instability

Several factors are involved in the regulation of physiological R-loop formation and resolution for review see references [60]. Our findings indicate decreased levels of important transcripts of *RNase HII* enzyme, which removes transcripts from the genome before DNA replication. The reduced levels of *RNase HII* transcripts in the skin of patients with psoriasis explain the higher levels of accumulation of lncRNA with DNA. At the same time, this indicates dysregulation of *RNase HII* transcripts, but we do not know if initially, this is due to a general transcription alteration. Down-regulation at the transcript level could well occur with deregulation of small non-coding RNAs at post-transcriptional levels or with a silencing event at the start of transcription. *RNase HII* transcription analysis in patients with psoriasis compared to control could evidence the controlling factors.

### 4.4. Psoriasis Patients and Cancer Risk

The possibility of cancer risk in patients with psoriasis is mentioned in the literature. Lymphomas and keratinocyte cancer have been reported to be at increased risk in patients with psoriasis [61,62]. However, it is not yet clear whether this is due in patients with psoriasis to repeated lesions conditions with chronic low-grade inflammation or to the result of repeated treatments. This could be due as generally proposed to the side effects of biological therapy on the innate and adaptive immune pathways. However, here results suggest that psoriasis genome is already engaged in telomere shortening with increased levels of TERRA attached to telomeres. In fact, we report that patients with psoriasis show already in normal skin higher levels of TERRA retention at the telomeres. It seems TERRA is recruited mostly on short telomere. Shortened telomeres are almost always an indicator of the disease. The shortening of TL in lesional skin is evident, but we do not yet know the TL after healing at the lesion points of the skin. At the repeated site of lesions after healing with or without therapy telomere length and genetic determination requires further studies. Environmental factors, like exposure to carcinogens, diet, tobacco, behavioral habit, and lifestyle, from early in life in the individual may affect this, as well as gene expression, and the same type of studies now can be developed in the model organisms.

Finally, we observe also more retention of abundant non-telomeric TERRA transcripts on the DNA, which confirms overall changes in RNA retention on the genome in patients with psoriasis.

### 4.5. For Patients with Psoriasis

Psoriasis is not yet a curable disease, and current treatments block intermediary molecules that play a role in the mechanism of inflammation or reduce the severity of the disease by suppressing or reducing the proliferation of keratinocytes. Psoriasis seriously and negatively affects the quality of life of patients. Psoriasis is consistently considered an inflammatory disease and, at this stage, the causality is uncertain. With our results, it appears that the epigenetic mark is already present in unaffected tissues of patients with psoriasis. It is the first time that it has been revealed with epigenetic signals in the form of lncRNAs marking the genome in patients with psoriasis. On the other hand, in psoriasis cytokine pathways are mainly described and are related to immune diseases. We are opening the question of causality in the development of inflammation due to epigenomic changes in the regulation of the factors involved in genome integrity such as RNaseHII.

Thus, there can be epigenetic mechanisms at play, and it is clearly very important to develop new strategies for the treatment of this disease [2].

In conclusion, the levels of TERRA and centromeric regions were higher in the RNA attached to the DNA of patients with psoriasis, regardless of clinical stage, thus making the complex lncRNA as a potential epigenetic DRNA marker and a therapeutic target.

## Figures and Tables

**Figure 1 biomolecules-12-00368-f001:**
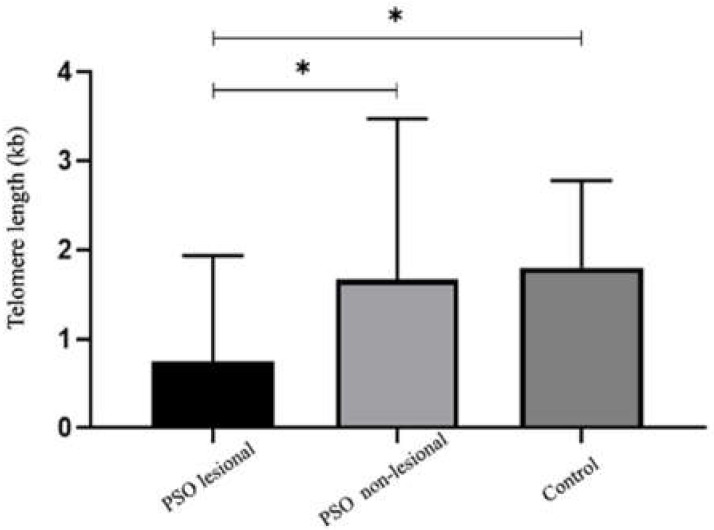
Telomere length in healthy skin tissues compared to psoriasis lesional and non-lesional tissues. Telomere length (TL) were determined by qPCR and TL of skin tissues in healthy control compared to PSO lesional and PSO non-lesional tissue (* *p* < 0.05). Telomere lengths are significantly shorter in PSO lesional tissue compared to healthy control tissue (* *p* < 0.05) PSO: Psoriasis, kb: kilobases.

**Figure 2 biomolecules-12-00368-f002:**
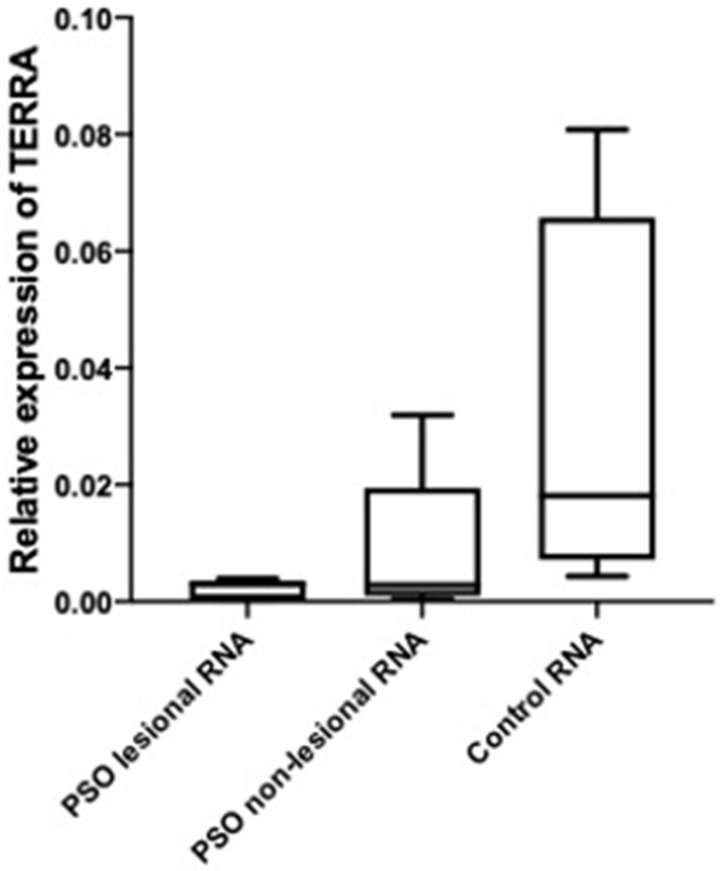
TERRA transcript levels in psoriasis and healthy control in terms of total free RNAs. The level of TERRA transcripts was determined by RT-qPCR analysis: relative normalized expression of TERRA in PSO lesional, non-lesional, and healthy control in terms of RNA (total free-RNA fraction) of total TERRA. No difference in TERRA level were found in psoriasis samples (*n* = 20) and control (*n* = 15).

**Figure 3 biomolecules-12-00368-f003:**
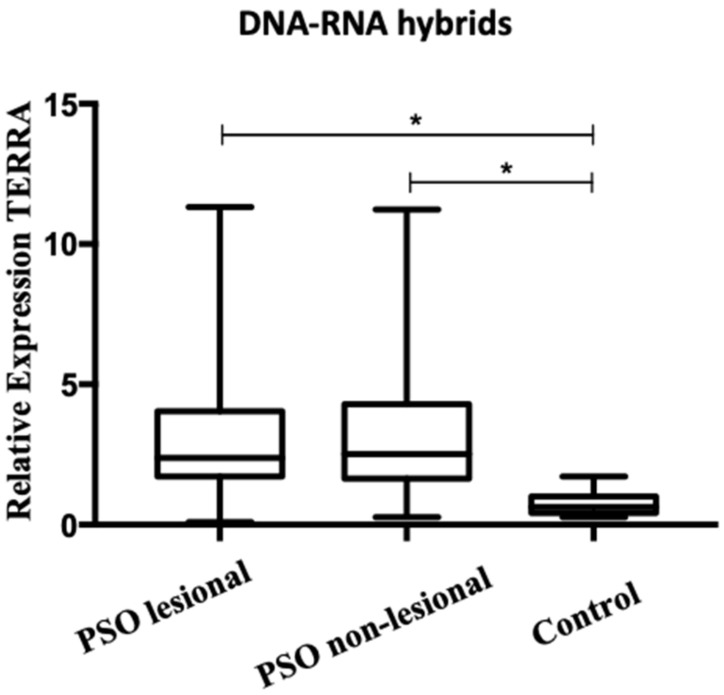
TERRA levels in DNA–RNA hybrids fraction of psoriasis and healthy control skin. The level of TERRA transcripts was determined by RT-qPCR analysis. Relative normalized level of TERRA expression from different chromosomes of psoriasis and healthy control of DNA–RNA hybrid fraction: Significant differences in the level of TERRA were found in psoriasis samples and control (* *p* < 0.05).

**Figure 4 biomolecules-12-00368-f004:**
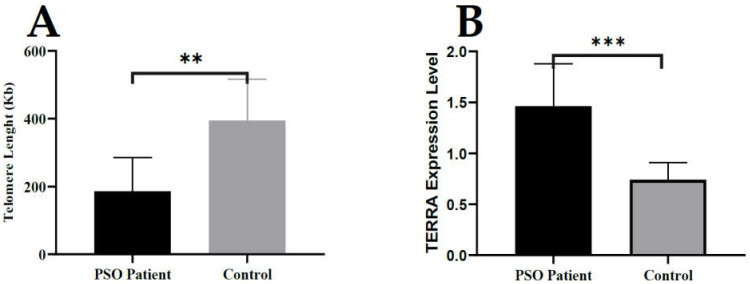
Telomere length and TERRA levels in blood cells from psoriasis patients and controls. Levels of TL (telomeres length) and TERRA transcripts were determined by qPCR or RT-qPCR analysis: (**A**) TERRA expression level in DNA–RNA hybrid (*p* = 0.0004) and (**B**) telomere length in blood tissue (*p* = 0.061) (** *p* < 0.05, *** *p* < 0.001).

**Figure 5 biomolecules-12-00368-f005:**
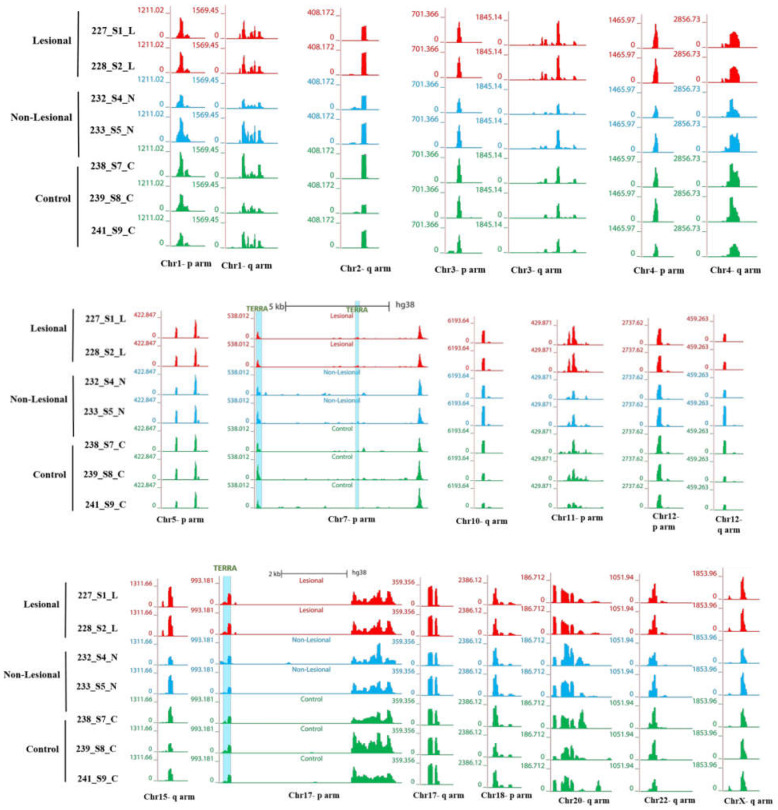
TERRA expression profiling genome-wide of DNA-bound RNA molecules over p-arm and q-arm chromosome telomeric regions. The Genome Browser figures show expression signal tracks of human DNA-bound RNA skin samples Lesional(L), non-lesional(N), and controls(C)) over telomeric regions of p-arm and q-arm chromosomes in GRCh38 assembly (minimum four consecutive TAACCC repeats) of telomeric regions. The red, blue, and green tracks show expression signals Telomeric TERRA repeats are overlapped to simple repeats. Blue lights are depicting TERRA regions specifically.

**Figure 6 biomolecules-12-00368-f006:**
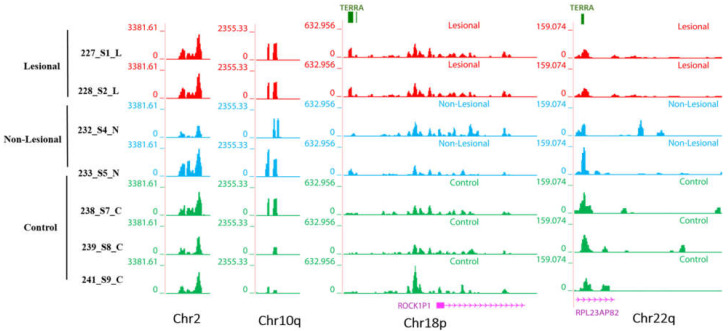
Non-telomeric region of TERRA sequences on chr2 (q31.3). The Genome Browser figures show read density tracks of human DNA-bound RNA skin samples (L, N, and C) over non-telomeric region of TERRA repeats on chromosomes 2, 10, 18, and 22. The tiny green bar is the location of TERRA sequences in GRCh38 genome assembly (minimum four consecutive TAACCC repeats) of telomeric regions. The annotation track is also displayed at the bottom of some graphs (GENCODE V36).

**Figure 7 biomolecules-12-00368-f007:**
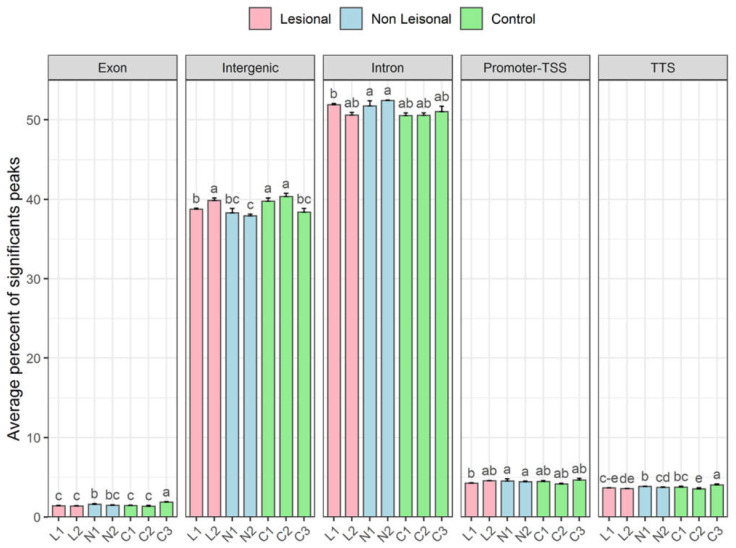
Peak finding. The average percentage of significant peaks of different genomic regions including exon, intergenic, intron, promoter-TSS, and TTS in all sample groups. Values are mean ± SD of each sample group biological replicates. Bar plots with different superscript are significantly different (one-way ANOVA) at the level of *p* < 0.05 followed by Duncan’s multiple comparison test. One-way ANOVA has been performed for each genomic region, respectively: (1L, 2L; lesional), (1N, 2N; non-lesional), (2C, 3C, and 5C; healthy controls). Each sample group including 2–3 biological replicates with four technical replicates.

**Figure 8 biomolecules-12-00368-f008:**
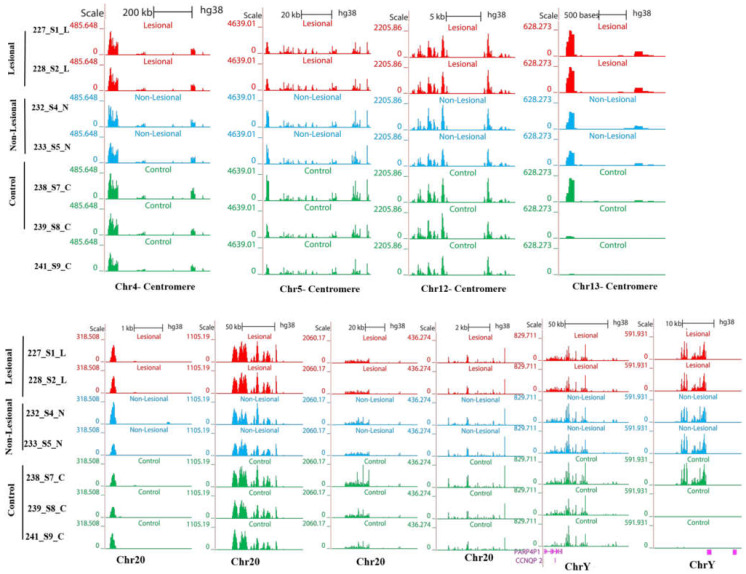
Centromeric and pericentric RNAs genome-wide profiling of DNA-bound RNA molecules. The browser of expression patterns of human DNA associated RNA samples over centromere regions of chromosomes 4, 5, 12, 13, 20, 21, and Y are shown at Figure 8. Centromeric signals are overlapped to simple repeats. A part of ChrY signals is overlapped to pseudogenes.

**Figure 9 biomolecules-12-00368-f009:**
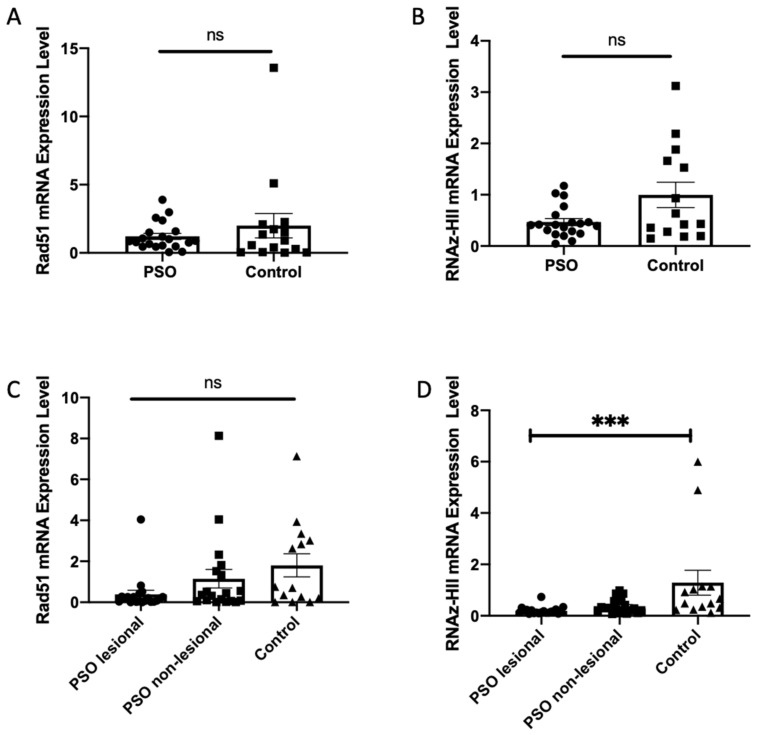
Comparison of *Rad51* and *RNase HII* transcripts expression level in blood samples from lesional and non-lesional tissues. Levels of transcripts were determined by RT-qPCR analysis. (**A**) *Rad51* expression level in blood samples compare with healthy control (*p* > 0.05). (**B**) *Rad51* expression level in lesional and non-lesional tissue compare with healthy control (*p* > 0.05). (**C**) *RNase HII* expression level in blood samples compare with healthy control (*p* > 0.05). (**D**) *RNase HII* expression level in lesional and non-lesional tissue compare with healthy control (PSO lesional vs. Control: *p* < 0.0001) (ns: nonsignificant, *** *p* < 0.001).

## Data Availability

To review GEO accession GSE188763: Go to https://www.ncbi.nlm.nih.gov/geo/query/acc.cgi?acc=GSE188763 (accessed on 24 January 2022). Enter token axyfuoyyjbyjlkt into the box.

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
