# Peer review of "Decrease in RNase HII and Accumulation of lncRNAs/DNA Hybrids: A Causal Implication in Psoriasis?"

_biomolecules, 2022, doi:10.3390/biom12030368_

Round 1

Reviewer 1 Report

In this manuscript, Mehmetbeyoglu and colleagues suggest a novel pathogenic mechanism associated to psoriasis, in which a decrease of RNase HII expression leads to the increase of lncRNAs in R-loop structures, in particular the telomere-associated TERRA lncRNA.

The main results are:

1 - the telomere length in lesional skin region of psoriasis patients is shorter compared to both non-lesional and healthy controls.

2 - A higher level of TERRA lncRNA is associated with telomeric DNA in psoriasis patients

3- in psoriasis patients, the levels of Ribonuclease HII transcript is decreased, suggesting a casual role in the increased levels of telomere-associated TERRA lncRNA

Overall, the research presented here shows some points of interest for the broad readership of Biomolecules. However, the manuscript is hard to read in its current version and some points should be addressed in order for it to be accepted.

My main concern is that I could not find a clear hypothesis on the association between the altered expression of RNAse HII and the accumulation of DNA/RNA hybrids. The authors should explain better the rationale of focussing only on this gene (and RAD51, that turned out not to be dysregulated). Also, from Fig. 9D seems that RNAse HII expression is reduced in both lesional and non-lesional tissue, while TERRA expression is affected only in lesional tissue (Fig. 3). Could the authors make some comments? Finally, the title looks misleading, since the potential association between RNAse HII and accumulation of lncRNAs/DNA hybrids was not the major point addressed by this manuscript, but just the last experiment shown.

Minor points:

As a general rule, the titles of paragraph in the “Results” section should address the main findings described; this is not the case, fro example, for sections 3.3.1, 3.3.2, 3.3.4.

Figures are too small and difficult to read. Figure layout should also be revised. Legends are duplicated

Please uniform the font and character size

Line 275: “Unlike the often used DRIP techniques…” please explain the “DRIP” acronym and include a reference

Author Response

Reviewer 1

Overall, the research presented here shows some points of interest for the broad readership of Biomolecules. However, the manuscript is hard to read in its current version and some points should be addressed in order for it to be accepted.

Authors:

We appreciate your time and help to improve and to make it accessible to larger scientists.

Reviewer 1

My main concern is that I could not find a clear hypothesis on the association between the altered expression of RNAse HII and the accumulation of DNA/RNA hybrids. The authors should explain better the rationale of focussing only on this gene (and RAD51, that turned out not to be dysregulated).

Authors:

Since that the genome is actively transcribed. RNA serves as a repair template (Chakraborty et al., 2016; Keskin et al., 2014), in DSB repair at transcription active regions; RNA forms a DNA-RNA hybrid to recruit repair factors (d’Adda di Fagagna, 2014), and triggers a transcription-associated (TA) homologous recombination repair (HRR) (Aymard et al., 2014; Ohle et al., 2016; Yasuhara et al., 2018). These pathways protect the actively transcribed regions and from genotoxic stresses.

RNases H are essential for degrading the RNA template during reverse transcription of the retroviral genome, thereby generating short RNA primers that initiate DNA synthesis (Flint et al., 2015).

Cellular RNase H enzymes share the common activity of degrading the RNA moiety of RNA-DNA hybrids necessary, for instance, to remove RNA primers during DNA replication.

We have added sentences to explain more clearly in Introduction and discussion.

Reviewer 1:

Also, from Fig. 9D seems that RNAse HII expression is reduced in both lesional and non-lesional tissue, while TERRA expression is affected only in lesional tissue (Fig. 3). Could the authors make some comments?

Authors:

TERRA expression is increased in both lesional and non-lesional tissue, please check again Figure 3 with higher resolution.

Reviewer 1:

Finally, the title looks misleading, since the potential association between RNAse HII and accumulation of lncRNAs/DNA hybrids was not the major point addressed by this manuscript, but just the last experiment shown.

Authors:

We are not agree, RNase HII is a major actor in resolving DNA/RNA hybrid. R-loops accumulate due to the lower expression levels of RNase HII.

We have added sentences in Introduction.

RNases H can remove the RNA moiety and prevent deleterious DNA breaks. RNase H2 knockout mice are not viable, and mutations in either of the human genes can cause Aicardi-Goutières Syndrome, a severe inheritable neurodevelopmental disorder (Crow et al., 2006). In this disease, uncleaved RNA-DNA hybrids accumulate within cells (Mackenzie et al., 2016).

Here, we show that DNA/RNA hybrids are accumulated because RNase HII is reduced. 

Reviewer 1:

Minor points:

As a general rule, the titles of paragraph in the “Results” section should address the main findings described; this is not the case, fro example, for sections 3.3.1, 3.3.2, 3.3.4.

Authors:

Exactly, sorry we removed.

Reviewer 1:

Figures are too small and difficult to read. Figure layout should also be revised. Legends are duplicated. Please uniform the font and character size.

Authors:

That’ is right we changed.

Reviewer 1:

Line 275: “Unlike the often used DRIP techniques…” please explain the “DRIP” acronym and include a reference

Authors:

We have added definition.

DRIP-seq (DRIP-sequencing) is a technology for genome-wide profiling of DNA-RNA hybrid called "R-loop". DRIP-seq techniques utilize antibody for DNA-RNA immunoprecipitation and DNA strand sequencing.

Reviewer 2 Report

The manuscript by Mehmetbeyoglu et al. presents interesting findings showing that TERRA levels were higher in patients with psoriasis. The manuscript is well written, direct to the point, and well presented. In addition, the authors have done a great job in introducing the topic to the reader, clearly explaining the rationale and the hypothesis. Except for a few issues, this reviewer believes that the study was well-conceived and properly analyzed, and the conclusions are supported by the data.

Major issues:

-Line 117  “consistent with age and gender as a control group “. Please explain it better and provide age and sex distributions. Please state if there were statistical differences, the test used for this calculation. If there were differences, explain how this was adjusted/addressed. I also assume that the biological variable is “sex”, not “gender”, please correct.

-Please also state how the authors addressed other possible variables that affect gene expression, such as the use of medications, tobacco, ancestry, pregnancy, etc.

Minor issues

-Introduction: several abbreviations were not introduced in full, including some protein names (e.g. TIN2, TPP1) and lncRNA (the last one was only introduced in the abstract”

The last two sentences of the introduction start very similarly “our results reveal”. I suggest rephrasing.

Author Response

Reviewer 2

Major issues:

-Line 117  “consistent with age and gender as a control group “. Please explain it better and provide age and sex distributions. Please state if there were statistical differences, the test used for this calculation. If there were differences, explain how this was adjusted/addressed. I also assume that the biological variable is “sex”, not “gender”, please correct.

Authors:

It is specified in results section as below:

Twenty patients with chronic plaque psoriasis were included in the study. Ten of these patients were women and ten were men. The healthy group consisted of 15 people, eight women (53.33%) and seven men (46.67%). Three sample groups of lesional (LA) and non-lesional (NL) psoriasis and healthy control were analyzed. The mean age of the control groups was 42.6 and 39.5 years, respectively. There is no statistically significant difference between the groups in terms of age and sex (p> 0.05). The mean of Psoriasis Area and Severity Index (PASI) of the patient group was 15.6.

And we added in methods:

Ten women and ten men patients with psoriasis were with fifteen healthy group, (eight women and seven men). The mean age of the control groups was 42.6 and 39.5 years, respectively. There is no statistically significant difference between the groups in terms of age and sex (p> 0.05).

Reviewer 2:

-Please also state how the authors addressed other possible variables that affect gene expression, such as the use of medications, tobacco, ancestry, pregnancy, etc.

 Authors:

We added a sentence in discussion. Those factors are reported to affect psoriasis, could be, but we did not addressed yet.

Reviewer 2:

Minor issues

-Introduction: several abbreviations were not introduced in full, including some protein names (e.g. TIN2, TPP1) and lncRNA (the last one was only introduced in the abstract”

Authors:

We added full name of proteins involved in telomere protection.

Reviewer 2:

The last two sentences of the introduction start very similarly “our results reveal”. I suggest rephrasing.

Authors:

Thanks for suggestion. That’ s right we changed.

Round 2

Reviewer 1 Report

The authors fully addressed my comments, therefore I recommend this manuscript for publication on Biomolecules